# Simple Donor–π–Acceptor Compounds Exhibiting Aggregation-Induced Emission as Hidden Fingerprints Detecting Agents

**DOI:** 10.3390/molecules28227597

**Published:** 2023-11-14

**Authors:** Patrycja Filipek, Hubert Hellwig, Agata Szlapa-Kula, Michał Filapek

**Affiliations:** 1Institute of Chemistry, Faculty of Mathematics, Physics and Chemistry, University of Silesia, Szkolna 9, 40-007 Katowice, Poland; patrycja.filipek@us.edu.pl (P.F.); agata.szlapa-kula@us.edu.pl (A.S.-K.); 2Center for Integrated Technology and Organic Synthesis, Research Unit MolSys, University of Liège, B-4000 Liège, Sart Tilman, Belgium; hellwig.hub@gmail.com

**Keywords:** aggregation-induced emission (AIE), latent fingerprints, donor–π–acceptor, electrochemistry, luminescence

## Abstract

Latent fingerprints are a significant carrier of information for a court expert. To detect this type of forensic trace, what is necessary is a method that is easy to use, compact, and versatile. The research aimed to investigate the physicochemical properties of luminescent substances of donor–π–acceptor systems in terms of their potential use in detecting hidden fingerprints. During the research, a group of fluorene compounds consisting of the (-CH=C(CN)(COOR)) moiety was designed and successfully synthesized. The optical, electrochemical, and aggregation-induced emission properties were studied. The aggregation-induced emission of compounds has been studied in the mixture of THF (as a good solvent) and water (as a poor solvent) with different water fractions ranging from 0% to 99%. Due to the molecular structure, substances showed different affinities to organic traces. As a result, it was noticed that all compounds showed the AIE phenomenon, while during tests on latent fingerprints, it was observed that two substances had particularly forward-looking features in this field.

## 1. Introduction

Latent fingerprints (LFP) are a significant information carrier. Fingerprints are one of the strongest pieces of evidence in lawsuits. This is due to their uniqueness; each person has a unique fingerprint. One could say that fingerprints are a unique code assigned to each person [1]. Their detection is constantly evolving, and new techniques are still being developed while the existing ones are still being improved [2]. Fingerprints have been used as proof of identification for over 100 years. Initially, it was based on evidence in the form of traces already embedded in substances [3]. Many methods of fingerprint detection have been developed over the years, but a significant number of traces remain undetected, or the obtained image quality is poor [4,5]. Even though fingerprints are an individual feature of each person, to verify/compare two fingerprints, it was necessary to determine which elements would be taken into account to obtain a reliable result [6]. Based on these characteristic elements of each fingerprint, a system was created that allowed the visual appearance of the sign to be transformed into a numerical value. This method detects and counts minutiae, i.e., elements of a fingerprint such as edges or forks. The first tier reveals the general flow of papillary folds. The second level is that the minutiae are known. The third level features are the shape of each ridge and pores [7,8]. These techniques can work in many ways. Some react with specific chemical components of LFP, while others take advantage of the differences between the composition of the print and that of the substrate (the substrate can be divided into three types: porous, semi-porous, and non-porous) [3]. Previous techniques, such as powder dusting, ninhydrin spraying, iodine, or cyanoacrylate smoking, face many challenges, such as low sensitivity or contrast. Therefore, it is required to develop a technique that is easy to apply, fast, and compact [9].

The powder method is one of the most widely used methods. The procedure is based on the visual contrast between the background surface and the powder-coated area of the fingerprint. The popularity of this method comes from the broad spectrum of powder application possibilities and their high efficiency. The effectiveness of this method depends on the visual contrast between the background surface and the area of the fingerprint that has been coated with the powder. More specifically, a key role in visualization is played by the adhesion forces between the powder and the fingerprint residue (this residue consists of eccrine sweat and sebaceous secretion). In contrast, these adhesive forces are influenced by two factors: pressure deficit and electrostatic attraction [10,11,12,13].

The ninhydrin method has been implemented to detect latent fingerprints since the late 1960s. Its primary purpose was porous surfaces [14]. Ninhydrin staining occurs when the ninhydrin molecule binds to the amino acid contained in the embedded fingerprint, which turns purple. A similar mechanism accompanies the iodine vapor method. The iodine vapor reveals a latent fingerprint as the sublimated iodine vapor dissolves into the fat or sweat contained in the LFP [15].

Cyanoacrylate glue smoking is another visualization technique for latent fingerprints. It involves exposing the material (which we believe has hidden fingerprints) to cyanoacrylate (otherwise known as super glue) at elevated temperature and humidity in a closed chamber. In the presence of glue, a polymer forms on the surface of the fingerprint stripes, making them visible [16,17].

A. Hussain Malik et al. focused their attention on the use of the aggregation-enhanced emission phenomenon. The developed CPE (3,3′-((2-(4-(1,2-diphenyl-2-(p-tolyl)vinyl)phenyl)-7-(7-methylbenzo[c][1,2,5]thiadiazol-4-yl)-9H-fluorene-9,9-diyl)bis(hexane-6,1-diyl))bis(1-methyl-1H-imidazol-3-ium) bromide) copolymer based on AIE (aggregation-induced emission) showed a strong emission on a solid substrate, thus obtaining good contrast and a high-resolution image. Moreover, the developed trace did not require any pre-treatment [18]. Another equally exciting way of using the AIE phenomenon was presented in the work by B. Zhang Tang and his group. They used silole compounds to develop LFP on wet and non-porous surfaces. As a result, it was observed that the resulting aggregates adhered to the material in the form of fingerprints and then strongly transformed UV light into visible light, thus allowing for good visualization of traces [19,20,21].

One example of AIE-based organic dye is the imidazole luminogen, whose core is fluorene. The molecule of such a compound showed strong fluorescence. This allows obtaining a clear and sharp image (many trace features are visible at different levels). Moreover, it is possible to apply such a luminogen to various surfaces and methods [22,23,24].

Besides the fact that a good contrast is obtained, an exciting aspect is the applying method of a given substance to the LFP. Classical methods do not positively affect the person’s health conducting the test or may destroy sensitive material [25]. Therefore, the much better solution turns out to be carrying the substance with an atomizer [26]. Y. Wang et al. used both the spraying and soaking methods, and both methods gave positive results on a variety of substrates except a paper-like substrate due to the characteristics of the substrate [27]. It is worth noting that it is not worth completely abandoning classic techniques such as powdering [28,29]. K. Singh and others presented an unconventional approach to the powder method. Namely, silica gel proved to be a substitute for fluorescent powder. The obtained results are comparable with the results for “classic” powders [30].

Sometimes, securing such a detected fingerprint and strengthening its image is necessary [31]. Smoking with cyanoacrylate glue (so-called super glue) is one of the chemical methods for detecting LFP. It is used on non-porous surfaces but also on glass and plastic [32]. The method is based on a polymerized cyanoacrylate ester deposition on latent fingerprint residues [33]. This method allows us to obtain unmistakable and stable fingerprints. Moreover, it is an effective and non-destructive method. In addition, this method can be further modified [34]. 

If the contrast is too low after the fuming process, it is necessary to use fluorescent staining [35]. This process can be performed in two stages or one step [36]. The one-step approach is more challenging because phosphors must be developed to be sublimated as standard smoke [37]. The universal approach is the two-stage approach, where the first stage is fuming, followed by reinforcement by applying a fluorescent powder [38]. Of course, there are many potential materials with potential properties for detecting hidden fingerprints: inorganic, hybrid organic–inorganic systems, and pure organic ones [39,40]. However, the most promising seem to be low-molecular-weight organic compounds triggering the AIE phenomenon [41]. AIE-based compounds are easy to obtain with good yield on a large scale compared to other compounds. Due to their high PL efficiency, the amount of the assay reagent needed to reveal LFP is minimal. They can also be applied to the tested surface as a powder and a solution (dropped or sprayed). Moreover, as in the case of the compound presented in this paper, they may contain lipophilic fragments, increasing the affinity for the detected traces.

## 2. Results and Discussion

The main idea of the research was to design, synthesize, and study the photophysical properties of compounds with potential applications to reveal LFP. According to the predictions (based on studies on structurally similar compounds), they should exhibit a high AIE index (a low quantum yield in solution and high as solid), having approximately identical electronic parameters (e.g., energy band gaps) but differing in lipophilicity and intermolecular interactions. The last feature should be provided by changing the carbon chain in the ester group. On the other hand, to ensure appropriate optical parameters (maximum absorption and emission bands), the molecules have a donor–π–acceptor architecture. Finally, the compounds should be obtained from cheap, readily available substrates in a simple manner with high yields. Considering the above assumptions, we have designed, obtained, and evaluated compounds with structures presented in Figure 1 (below).

### 2.1. DFT Calculations

In the first step, DFT calculations were performed to check the position of the frontier molecular orbitals (together with their energy), IP, EA, and band gap of the investigated compounds. B3LYP/6-311+G was used as the functional and basis. The calculations were carried out in the Gaussian16 program [42] with acetonitrile as a solvent.

The discussed molecules are based on the donor–acceptor (D–A) structure. First, it was checked whether the change of the acceptor group affected the values of the frontier molecular orbitals. In the case of the HOMO (highest-occupied molecular orbital), it is largely located on the fluorene moiety. Moreover, the energy values of this orbital range from −6.36 V to −6.39 V for all molecules. The LUMO (lowest-unoccupied molecular orbital) is mostly dominated by the 2-cyanoacrylic ester moiety. Interestingly, this orbital does not include the alkyl chains that change in these molecules and the energy values of the orbital differ little (−2.84 V to −2.91 V) (Table 1). This indicates a high similarity of the molecules and no influence of the 2-cyanoacrylic ester derivative motif on the presented compounds. This observation is consistent with structurally similar molecules [40]. In the next step, the values of IP, EA, and Eg(DFT) were calculated. These values more closely correspond to experimental values obtained from electrochemical studies. For all molecules, the IP values were almost the same (6.14–6.17 eV). It is surprising that in the case of EA, the values were also similar and were in the range of 3.16 to 3.22 eV. The energy gap of series M1–M4 was in the range of 2.95–2.99 eV, which is favorable for this type of compound (Table 2). Moreover, the reorganization energies of holes and electrons were determined. When the IP(v) values were low and EA(v) were high, efficient load transfer could be achieved [43,44,45]. Analyzing the obtained results, it is clear that the EA(v) value is high. However, the IP(v) value is not low, which may hinder the effective transfer of electrons. All compounds are characterized by low λhole values. They also have similar λelectron values. Smaller values of the λhole relative to the λelectron suggest the easier transport of holes relative to electrons (see Table 2). Modifying the acceptor group did not cause significant changes in the width of the energy gap and the λ values of holes and electrons. Comparing compounds M1–M4 with their analogs reported before, we can observe that the fluorene substituent increases the energy gap in relation to the carbazole substituent [46].

### 2.2. Photophysical Characterization

In the next research stage, it was checked whether the molecules exhibited similarity in their electronic and physicochemical parameters in diluted solutions. To verify this hypothesis, several electrochemical measurements were conducted. Since we were interested in the values determined for isolated molecules, we studied them using the cyclic voltammetry (CV) method for dilute solutions (to avoid intermolecular interactions).

As can be seen from Figure 2, in general, the behavior of all tested compounds was similar, both in terms of the values (potentials) at which the electrode processes took place and in terms of their thermodynamical character. The Eox (oxidation peak onset) values in each case lay between 1.28 and 1.33 V (thus, the influence of the ester group substituent was negligible). In the case of Ered (reduction peak onset), a similar convergence of the values was observed (from −1.43 to −1.55). Due to CV, it was possible to determine the ionization potential (IP) and electron affinity (EA) of the investigated compounds. For the diluted solution, the above-mentioned values could be treated (approximately) as HOMO and LUMO levels (IP and EA, respectively). This, in turn, allowed us to calculate the energy band gap (Eg), which is the key value (Table 3). The Eg for voltamperometric measurements showed that all compounds were similar and in the range of 2.71 eV and 2.86 eV. This value is a perspective (from the potential application point of view) because (based on a mathematical formula) it is possible to predict the length of the absorbed wavelength. Using the Eg = 1240/λ formula, we could calculate that all compounds should absorb light between ultraviolet and visible. The relationship mentioned above assumes that the transfer of an electron from the HOMO to the LUMO requires only the amount of energy needed to overcome the Eg barrier. In other words, it was assumed that both orbitals were located on the same fragment of the molecule.

UV-Vis spectroscopy measurements confirmed the electrochemical predictions. All compounds exhibited similar behavior in diluted solutions (solid lines in Figure 3). They were characterized by one broad band laying at the end of the visible light region. However, the exact values of the peak maxima were closer to each other in the case of M1 and M2. On the other hand, the energy band was the highest for M3 (i.e., a molecule with tert-butyl moiety). Additionally, the molar absorption coefficient decreased gradually from M1 to M4. 

As one can see in Figure 3 (dashed lines), absorption bands of all compounds in the MeCN/water mixture are always blue-shifted (in reference to their solutions in pure MeCN, with the same concentration). MeCN serves as a “good solvent”, while water is (in this case) a “bad solvent”. Thus, in this mixture, the investigated compounds start to aggregate [47]. The blue-shifting of the bands suggests that for this type of compound, intermolecular interactions are energetically favorable—they stabilize excited states and make the HOMO to LUMO electron transition easier.

In the next step, the photoluminescence properties of the investigated compounds (in a diluted solution) were measured and evaluated. As shown in Appendix A, the excitation band (dotted lines) increases to higher wavelengths from M1 to M4, but surprisingly, the emission wavelength is the same for M1, M2, and M4 (496 nm). The only exception is M3, for which emission is slightly shifted to higher energy light—i.e., 485 nm. 

### 2.3. Aggregation-Induced Emission Investigations

As the introduction mentions, the quantum yield of such compounds in a solution is low (>2%). However, this changes significantly in the case of powders (Figure 4). A significant increase in quantum efficiency suggests the presence of AIE properties. The study of the properties of the compound in a mixture of good and bad solvents (with different ratios) is one of the most important studies that allowed the confirmation of the phenomenon of aggregation-enhanced emission. After preliminary tests, a mixture of THF (as a good solvent) and H_2_O (as a poor solvent) with different fractions of water (fw) in the range from 0% to 99% was selected. For compound M1 (in pure THF solution), almost no fluorescence was observed under a 366 nm UV lamp. In addition, no visible changes were observed when the water: THF ratio was below 60%. Above 70%, there was a significant increase in intensity. Surprisingly, the highest quantum yield (68%) was obtained when fw = 90%; for higher water content, it dropped to 47%. In comparison, compound M3 with a tert-butyl substituent behaved quite differently. No visible changes were observed for this compound when the water: THF ratio was below 80%. Just above 90%, a significant increase in emissions was detected. Thus, as can be seen in Figure 5, only the case of the M1 and M2 AIE has a high value. 

Differences in the behavior of molecules can also be seen by comparing their photoluminescence. As can be seen, the first two compounds show much higher quantum yields. Furthermore, molecules M1 and M2 begin to aggregate at lower water contents (Figure 6). This also explains the decrease in yield observed for M1 (Table 4); at higher water contents, the precipitation may be too fast for the substance to reach its optimal configuration. In addition, it may contain more defects and trapped water molecules, which may negatively affect its physicochemical properties.

The conducted research confirms the postulated research hypotheses regarding the influence of the molecule’s structure on the properties of AIE. All substances in a dilute solution show very similar PL parameters and quantum yields. However, their behavior changes significantly as the concentration increases, i.e., when the molecules aggregate. For the M1 derivative, the AIE feature is the highest, as expected. This molecule possesses only a small methyl group as a substituent in the ester group. This means that in a solid (and concentrated solution), two molecules exhibit strong intermolecular interactions (because they can achieve the most favorable spatial configuration), which stiffen the molecule. As it is a well-known fact, one of the critical parameters causing the AIE phenomenon is the limitation (freezing) of rotation around a single bond. Analyzing the electrochemical, DFT, and optical results, the bond between the fluorene and cyanoester fragments is stiffened (in this case). In turn, for the M3 molecule, the tert-butyl group is so spherically expanded that it prevents the molecule from having strong intermolecular interactions, so some energy is consumed for non-radiative transitions related to rotation.

### 2.4. Identification of Latent Fingerprints 

Concerning the scoring mentioned in Appendix A, it allows us to evaluate the obtained fingerprint image, which, in turn, translates into usefulness as evidence. Looking at the pictures above (Figure 7), it can be concluded that the compound M1 could receive the maximum number of points on a four-point scale because all details are visible and, above all, the entire fingerprint is recognizable. For the image obtained for the compound M2, three points could be awarded because the fingerprint is recognizable, but not all the pattern details are visible. The situation is similar for the M3 union. The lowest “score”—1 would obtain compound M4 because the obtained image does not show exact details; it can only be stated that there was contact with the ground of the finger.

## 3. Materials and Methods

### 3.1. General Methods

All chemicals and starting materials were commercially available and were used without further purification. Solvents were distilled per standard methods and purged with nitrogen before use. Unless otherwise indicated, all reactions and measurements were carried out under an argon atmosphere. Column chromatography was carried out on Merck Silica Gel 60. Thin-layer chromatography (TLC) was performed on silica gel (Merck TLC Silica Gel 60 F254). The 1H NMR and 13C NMR spectra were recorded using a Bruker Avance UltraShield 400 MHz spectrometer. The peaks were referenced to the residual CDCl3 (7.28 and 77.04 ppm) resonances in 1H and 13C NMR spectra, respectively. UV/Vis spectra were recorded with a Hewlett–Packard model 8453 UV/Vis spectrophotometer in MECN and THF solutions. Electrochemical measurements were carried out with an Eco Chemie Autolab PGSTAT128n potentiostat using glassy carbon (with diam. 2 mm) as the working electrode, while platinum coil and silver wire were used as auxiliary and reference electrodes, respectively. Potentials are referenced with respect to ferrocene (Fc), which was used as the internal standard. Cyclic and differential pulse voltammetry experiments were conducted under argon in a standard one-compartment cell in MeCN (Carlo Erba, HPLC grade). Bu_4_NPF_6_ (Aldrich; 0.2 M, 99%) was used as the supporting electrolyte. Emission spectra were recorded on an F-7000 HITACHII fluorescence spectrophotometer equipped with a Xenon lamp and Solid Sample Holder. Quantum yields Φ were obtained via the absolute method using an integrating sphere. The emission spectra were recorded by exciting the samples at their excitation spectral maxima unless otherwise mentioned. The quantum theoretical calculations were performed using density functional theory (DFT), with an exchange-correlation hybrid functional B3LYP and the basis 6-311+G for all atoms. The calculations were carried out with the use of the Gaussian 09 program. 

### 3.2. Synthesis and Characterization

#### 3.2.1. Synthesis of M1, methyl 2-cyano-3-(9H-fluoren-2-yl)prop-2-enoate

In a 25 mL flask fitted with a small Dean–Stark apparatus (pre-filled with toluene and shielded with aluminum foil) was placed 388 mg (2.00 mmol) of 9-H-fluorene-2-carboxaldehyde, 180 μL (202 mg, 2.04 mmol) of methyl cyanoacetate, 5 mL of toluene, 125 μL of AcOH, and 38 mg (0.49 mmol) of anhydrous ammonium acetate. The mixture was heated to reflux for 4–5 h. During heating, a bright yellow-green deposit appeared on the walls of the reaction flask. After cooling, the mixture was dissolved in DCM (with heating) and washed 3× with H_2_O. The DCM layer was dried over MgSO_4_, and the solvent was removed on a rotary evaporator. The crude product was recrystallized from hot trichloroethylene, yielding 210 mg (Yield: 38%) of bright yellow crystals. The 1H NMR (400 MHz, CDCl_3_) δ 8.34 (s, 1H), 8.31 (s, 1H), 7.99 (dd, *J* = 7.7, 1.5 Hz, 1H), 7.93–7.85 (m, 2H), 7.66–7.58 (m, 1H), 7.48–7.39 (m, 2H), 4.00 (s, 1H), and 3.97 (s, 1H). The 13C NMR (101 MHz, CDCl_3_) δ 163.37, 155.33, 147.26, 144.63, 144.01, 140.24, 131.33, 129.90, 128.58, 127.22, 127.18, 125.31, 121.03, 120.40, 115.91, 100.87, 53.15, and 36.87.

#### 3.2.2. Synthesis of M2, ethyl 2-cyano-3-(9H-fluoren-2-yl)prop-2-enoate

The compound was synthesized using the procedure applied for the synthesis of M1. After recrystallization from trichloroethylene, 196 mg (Yield: 34%) of bright yellow crystals were obtained. The 1H NMR (400 MHz, CDCl_3_) δ 8.33 (s, 1H), 8.30 (s, 1H), 7.99 (dd, *J* = 8.2, 1.9 Hz, 1H), 7.93–7.83 (m, 2H), 7.62 (dq, *J* = 6.5, 0.9 Hz, 1H), 7.50–7.38 (m, 2H), 4.42 (q, *J* = 7.1 Hz, 2H), 4.00 (s, 2H), and 1.44 (t, *J* = 7.1 Hz, 3H). The 13C NMR (101 MHz, CDCl_3_) δ 162.95, 155.26, 147.19, 144.65, 144.00, 140.24, 131.41, 129.90, 128.59, 127.24, 127.19, 125.36, 121.07, 120.45, 116.10, 101.10, 62.61, 36.90, and 14.22.

#### 3.2.3. Synthesis of M3, tert-butyl-2-cyano-3-(9H-fluoren-2-yl)prop-2-enoate

The compound was synthesized using the procedure applied for the synthesis of M1. The pure compound was isolated using column chromatography (SiO_2_, DCM:hexane, 3:1 vol.). After evaporation of the solvent, 192 mg (Yield: 30%) of a bright-yellow solid was obtained. The 1H NMR (400 MHz, CDCl_3_) δ 8.26 (s, 1H), 8.24 (s, 1H), 7.96 (dd, *J* = 8.1, 1.8 Hz, 1H), 7.90–7.84 (m, 2H), 7.64–7.59 (m, 1H), 7.48–7.39 (m, 2H), 3.98 (s, 2H), and 1.63 (s, 9H). The 13C NMR (101 MHz, CDCl_3_) δ 161.76, 154.31, 146.83, 144.58, 143.93, 140.30, 131.14, 130.03, 128.46, 127.20, 127.06, 125.33, 120.99, 120.39, 116.30, 102.83, 83.57, 36.90, 28.02, and 27.89.

#### 3.2.4. Synthesis of M4, 2-methoxyethyl 2-cyano-3-(9H-fluoren-2-yl)prop-2-enoate

The compound was synthesized using the procedure applied for the synthesis of M1. The pure compound was isolated using column chromatography (SiO_2_, DCM:hexane 2:1 vol.). After evaporation of the solvent, 230 mg (Yield: 36%) of a bright-yellow solid was obtained. The 1H NMR (400 MHz, CDCl_3_) δ 8.29 (s, 1H), 8.24 (s, 1H), 7.95 (dd, *J* = 8.1, 1.7 Hz, 1H), 7.87–7.81 (m, 2H), 7.59 (dd, *J* = 6.2, 2.2 Hz, 1H), 7.47–7.36 (m, 2H), 4.49 (t, *J* = 4.7 Hz, 2H), 3.94 (s, 2H), 3.75 (t, *J* = 4.7 Hz, 2H), and 3.47 (s, 3H). The 13C NMR (101 MHz, CDCl_3_) δ 162.94, 155.47, 147.23, 144.65, 143.97, 140.17, 131.41, 129.80, 128.62, 127.23, 127.22, 125.34, 121.07, 120.43, 115.92, 100.75, 70.13, 65.47, 59.19, and 36.85.

## 4. Conclusions

In this work, four fluorene core compounds were synthesized with good yields. A series of tests were then carried out. It was confirmed that the obtained molecules exhibited the AIE phenomenon (in an acetonitrile/water mixture). Subsequently, the energy parameters for the obtained compounds were checked (energy levels were determined using cyclic voltammetry, UV-Vis spectroscopy, and DFT calculations). Photoluminescence and UV-Vis spectroscopy measurements were also performed in dilute solutions and for solids (powders). In the case of M1 and M2, strong aggregation-induced emission was observed. Surprisingly, for M1, the highest quantum yield (68%) was obtained when fw = 90%, while for higher water content, it dropped to 47%. During investigations on latent fingerprint detection, it was observed that two substances had particularly forward-looking features in this field. Summing up, the presented synthesis strategy allows in a cheap, convenient, and large-scale way to obtain pure compounds with the expected photophysical properties, and thus their potential use for detecting latent fingerprints.

## Figures and Tables

**Figure 1 molecules-28-07597-f001:**
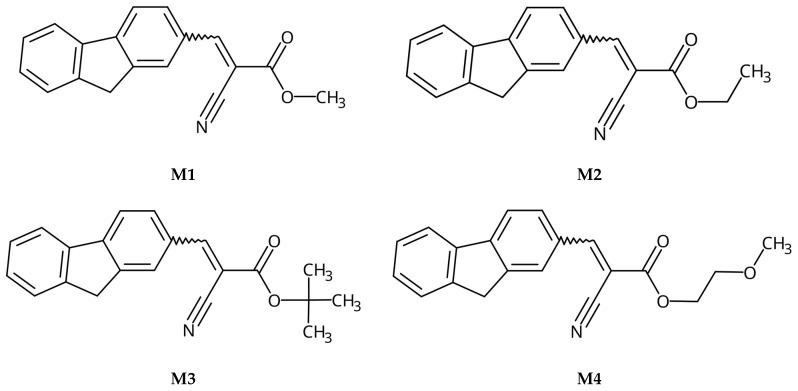
Structures of obtained compounds (**M1**–**M4**).

**Figure 2 molecules-28-07597-f002:**
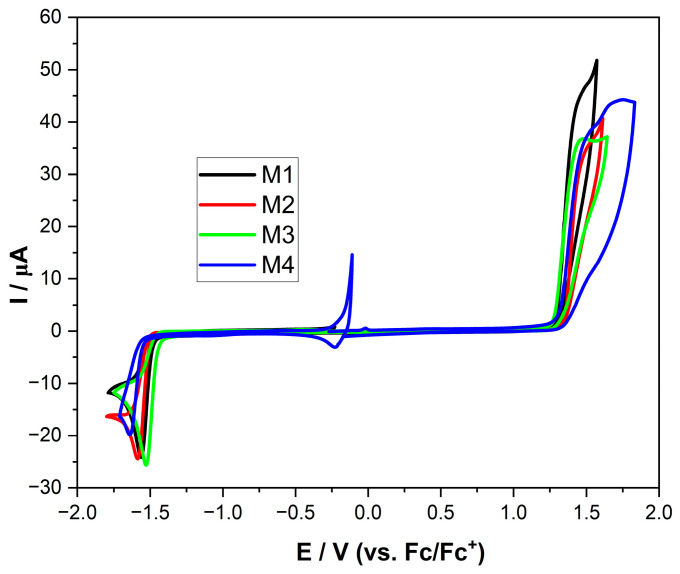
Cyclic voltammograms of the investigated compounds with sweep rate ν = 100 mV/s, 0.1 M Bu_4_NPF_6_ in MeCN.

**Figure 3 molecules-28-07597-f003:**
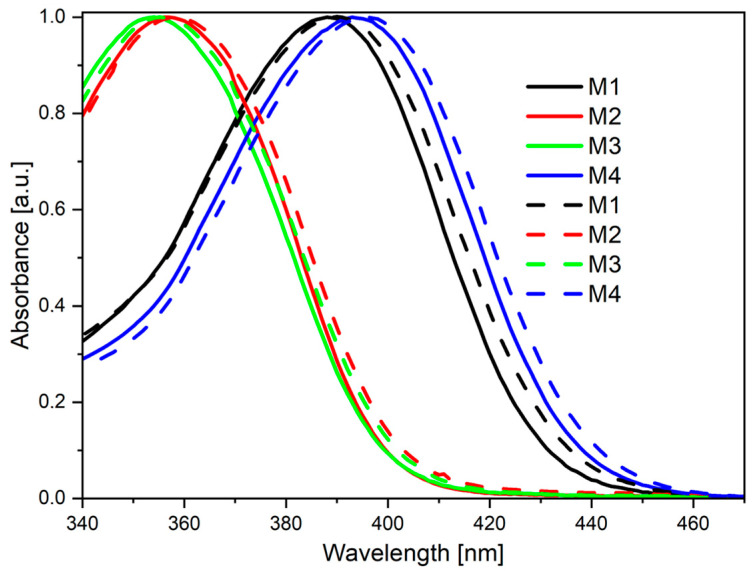
Normalized electronic absorption in MeCN c = 1 × 10^−5^ mol/L (solid lines) and MeCN/water mixture (c = 1 × 10^−5^ mol/L) 9:1*V*/*V* (dashed lines).

**Figure 4 molecules-28-07597-f004:**
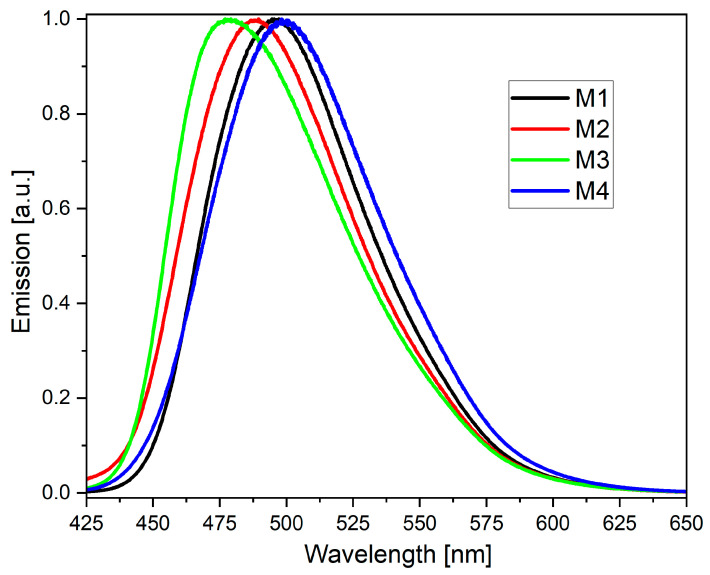
Normalized emission spectra of investigated compounds M1–M4 (for powders).

**Figure 5 molecules-28-07597-f005:**
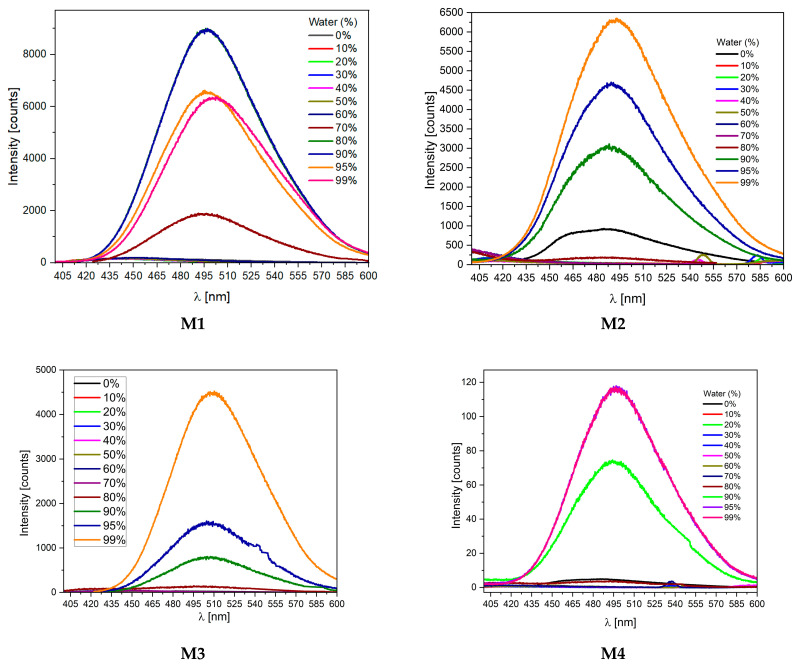
Differences in photoluminescence behavior during AIE investigations in THF:water mixtures for obtained compounds (**M1**–**M4**).

**Figure 6 molecules-28-07597-f006:**
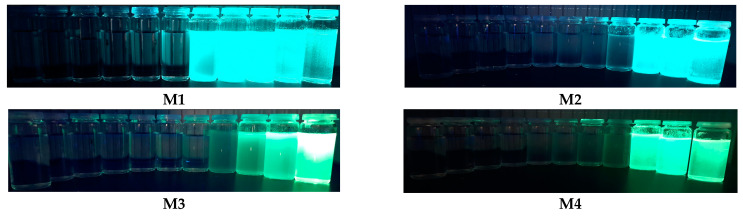
Photos taken during illumination by light with a 366 nm wavelength. In each column, the vessels are arranged to increase the water fraction in water: THF solutions (from left to right), i.e., I. 10%; II. 20%; III. 30%; IV. 40%; V. 50%; VI. 60%; VII. 70%; VIII. 80%; IX. 90%; X. 95%; and XI. 99%. **M1**–**M4** refers to the numbering of the obtained compounds.

**Figure 7 molecules-28-07597-f007:**
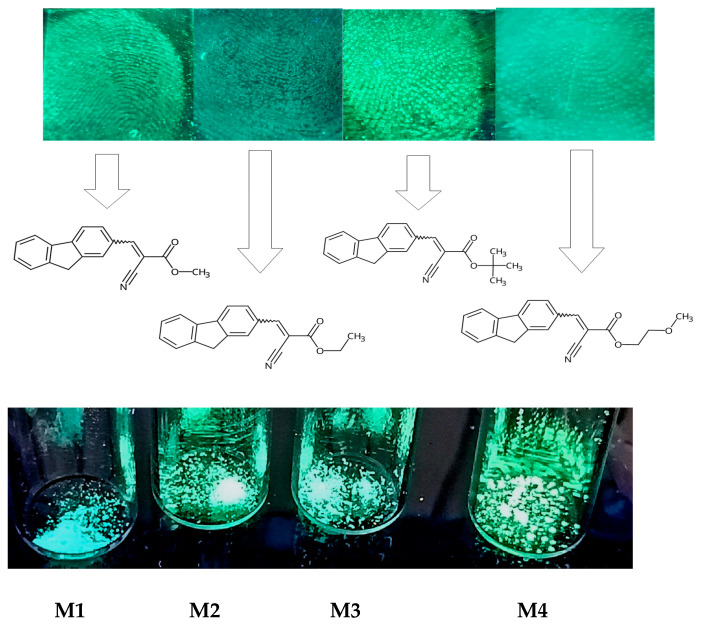
Pictures of investigated compounds from inside of the fluorescence spectrophotometer during irradiation (**top**) and picture of the investigated compounds (as a powder) during irradiation with 366 nm light (**bottom**). Samples are arranged in order from M1 to M4.

**Table 1 molecules-28-07597-t001:** The HOMO and LUMO levels together with calculated bond lengths.

CODE	HOMO	LUMO	BOND LENGTHS
1	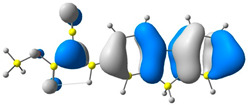	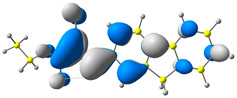	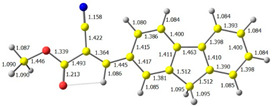
	E = −6.39 eV	E = −2.91 eV	
2	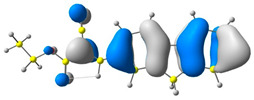	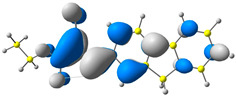	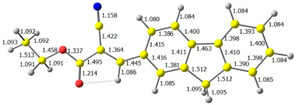
	E = −6.38 eV	E = −2.89 eV	
3	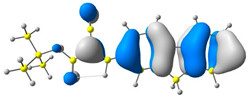	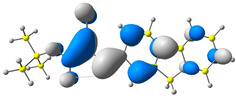	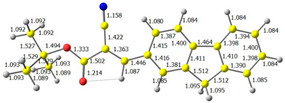
	E = −6.36 eV	E = −2.84 eV	
4	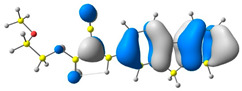	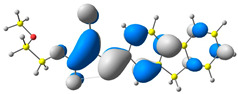	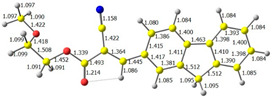
	E = −6.39 eV	E = −2.90eV	

**Table 2 molecules-28-07597-t002:** Calculated ionization potentials and electron affinities (vertical and adiabatic), energy gap, as well as hole and electron reorganization energies and extraction potentials for **1**–**4**.

Code	IP(v) [eV]	IP(a) [eV]	EA(v) [eV]	EA(a) [eV]	λ_hole_ [eV]	λ_electron_ [eV]	HEP [eV]	EEP [eV]
M1	6.27	6.17	3.05	3.22	0.20	0.34	6.07	3.39
M2	6.26	6.16	3.03	3.21	0.20	0.34	6.06	3.37
M3	6.24	6.14	2.98	3.15	0.20	0.35	6.04	3.33
M4	6.27	6.17	3.04	3.22	0.19	0.34	6.07	3.39

EEP = E^0^(M^−^) − E^−^(M^−^); HEP = E^+^(M^+^) − E^0^(M^+^); λ_electron_ = EEp − EA_v_; λ_hole_ = IP_v_ − HEP.

**Table 3 molecules-28-07597-t003:** IP, EA, and energy gap values for M1–M4.

Code	Eox	Ered	IP ^a^	EA ^a^	Eg (cv) ^a^	λonset	Eg (Opt) ^b^	Stokes Shift [nm]	ɛ [L/mol × cm]
1	1.28	−1.47	−6.38	−3.63	2.75	435	2.85	97	110,000
2	1.33	−1.49	−6.43	−3.61	2.82	442	2.80	139	89,000
3	1.28	−1.43	−6.38	−3.67	2.71	400	3.10	144	80,000
4	1.31	−1.55	−6.41	−3.55	2.86	402	3.08	104	45,000

^a^ calculated from CV measurements (IP = −5.1 − Eox; EA = −5.1 − Ered; Eg(CV) = Eox (onset) − Ered (onset)); ^b^ obtained from UV-Vis using Eg = 1240/λ_onset_ formula; ε = A/c × l.

**Table 4 molecules-28-07597-t004:** Photophysical parameters obtained from UV-Vis spectroscopy and photoluminescence measurements.

Code	UV-Vis λ_max_ [nm]	PL λ_max_ [nm]
M1 THF ^a^	388	484 (>1%) ^c^
M1 THF:water ^b^	390	499 (68%) ^c^
M1 POWDER	389	495 (47%) ^c^
M2 THF ^a^	357	496 (1%) ^c^
M2 THF:water ^b^	358	442 (23%) ^c^
M2 POWDER	357	489 (46%) ^c^
M3 THF ^a^	354	497 (>1%) ^c^
M3 THF:water ^b^	357	454 (4%) ^c^
M3 POWDER	355	479 (34%) ^c^
M4 THF ^a^	393	497 (>1%) ^c^
M4 THF:water ^b^	397	509 (1%) ^c^
M4 POWDER	398	498 (2%) ^c^

^a^ (c = 1 × 10^−5^ mol/L), ^b^ (c = 1 × 10^−5^ mol/L) 9 water: 1 THF (*V*/*V*); ^c^ quantum yield.

## Data Availability

Data are contained within the article.

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
