# Peer review of "Simple Donor–π–Acceptor Compounds Exhibiting Aggregation-Induced Emission as Hidden Fingerprints Detecting Agents"

_molecules, 2023, doi:10.3390/molecules28227597_

Round 1
Reviewer 1 Report
Comments and Suggestions for Authors
In this work, Patrycja Filipek et al. reported the “Simple donor-π-acceptor compounds exhibiting aggregation-induced emission as hidden fingerprints detecting agents”. the authors synthesized four fluorene core compounds with high yields. Subsequent tests confirmed that these molecules exhibited Aggregation-Induced Emission (AIE) behavior in an acetonitrile/water mixture. The energy parameters of these compounds were thoroughly analyzed, including their energy levels determined through cyclic voltammetry, UV-VIS spectroscopy, and DFT calculations. Photoluminescence and UV-VIS spectroscopy measurements were conducted both in dilute solution and as solid powders. Notably, two of the compounds, M1 and M2, displayed strong aggregation-induced emission properties. Additionally, the study explored the application of these compounds in latent fingerprint detection, revealing their promising potential for this field. Based on these findings, the manuscript should be considered for publication in “Molecules” upon addressing following issues.
1. Can AIE Molecules offer any advantages in Fingerprint Detection applications? It appears that organic molecular systems may be susceptible to environmental influences. In the introduction section, the authors should engage in a comparative discussion with respect to inorganic systems (Nanomaterials, 2022, 12; 400; Inorganic Chemistry Communications, 2023, 155: 111133), organic systems (Adv. Funct. Mater., 2019, 29, 1807599), and organic-inorganic hybrids (Research, 2021, 9862327; Inorg. Chem. Front., 2021, 8, 4924), highlighting the strengths of the AIE system.
2. The author discusses the frontier orbitals and suggests that these molecules may belong to the D-A system. However, based on the distribution maps, it appears that the nature of the D-A system is not clearly evident. Therefore, the author is encouraged to calculate the electron-hole distribution maps and furnish additional theoretical evidence for clarification.
3. All abbreviations should be noted when they first appear.
4. In Figure 3, some of the curves appear to be incomplete. Could you please provide additional information or make necessary revisions to ensure clarity?
5. The introduction in section 3.4 seems a bit redundant and can be placed in SI.
6. In Fig. 6, a scale bar is needed.
7. The differences in optical behavior between the reported cases require further discussion.
Comments on the Quality of English LanguageMinor editing of English language required.
Reviewer 2 Report
Comments and Suggestions for Authors
1. Line 213-214 mentioned “As shown in Figure 3, the excitation band (dotted lines) increases to.......”. However, I can not find the excitation band in Figure 3.
2. Two "Figure 3"s appear in the paper, the latter one should be Figure 4? Moreover, which part in the paper discuss the result of “figure 4”? Section 2.3.?
3. The highest quantum yield (68%) were obtained when fw=90%, it is a good result. However, considering there are some different ways to measure the QY, the used method, experiment and basic data in this paper should be provided in detail.
4. In the paper and its SI, I did not find any account for the machine type and operation procedure of the fluorescent measurement. In particular, the method to measure fluorescent data of powder samples should be mentioned. Moreover, it is better to provided a fluorescent picture of powder samples under UV light, to concrete the emission results in solid state (powder).
5. A different reagent (MeCN) has been used in Table 5, why there is no mention or explanation in the discussion part.
6. There are too much introduction of basic knowledge about latent fingerprint from Line 261 to Line 294, which should not be appeared in discussion section. Similarly, Table 6 is better moved to SI.
7. The four compounds M1 to M4 showed significantly different properties in AIE behavior. Thus, an important question rise, their properties had any relationship to their molecular structure? In my option, this is the novelty of this paper. Therefore, some discussion concerning mechanism and structure-to-property expatiation should be added in this paper.
Round 2
Reviewer 1 Report
Comments and Suggestions for Authors
The author has addressed the previous issues very well. I have no further questions at this time.
Reviewer 2 Report
Comments and Suggestions for Authors
In my opinion, the revised manuscript is ok and can be accepted.